# Knowledge and Perceptions of Molar Incisor Hypomineralisation among General Dental Practitioners, Paediatric Dentists, and Other Dental Specialists in Indonesia

**DOI:** 10.3390/dj10100190

**Published:** 2022-10-12

**Authors:** Enrita Dian, Sarworini Bagio Budiardjo, Aghareed Ghanim, Lisa Rinanda Amir, Diah Ayu Maharani

**Affiliations:** 1Faculty of Dentistry, Universitas Indonesia, Jakarta 10430, Indonesia; 2Department of Paediatric Dentistry, Faculty of Dentistry, Trisakti University, Jakarta 11440, Indonesia; 3Department of Pedodontics, Faculty of Dentistry, Universitas Indonesia, Jakarta 10430, Indonesia; 4Faculty of Medicine Dentistry and Health Sciences, University of Melbourne, Melbourne, VIC 3010, Australia; 5Department of Oral Biology, Faculty of Dentistry, Universitas Indonesia, Jakarta 10430, Indonesia; 6Department of Preventive and Public Health Dentistry, Faculty of Dentistry, Universitas Indonesia, Jakarta 10430, Indonesia

**Keywords:** knowledge, perceptions, molar incisor hypomineralisation, dental practitioners

## Abstract

Molar incisor hypomineralisation (MIH) is a qualitative, dental enamel hypomineralisation defect affecting one to four first permanent molars, characterised by the present of asymmetries demarcated opacities, and is prevalent worldwide. An early diagnosis of MIH is crucial, to prevent further complications including the development of dental caries, hypersensitivity, pulp inflammation, and pain. Therefore, a thorough understanding of MIH is of clinical importance. This cross-sectional study aimed to evaluate the knowledge and perception of MIH among general dental practitioners (GD), paediatric dentists (PD), and other dental specialists (DS) in Indonesia using a self-administered questionnaire. Chi-square tests and Kruskal–Wallis tests were employed to analyse the data. A total of 266 dental practitioners—112 GD, 84 PD, and 60 DS—were included in this study and completed the online questionnaire. There were significant differences in the overall knowledge scores between GD, PD, and DS (*p* < 0.001). Specifically, the different between the tested groups were observed in the knowledge of caries pattern related to MIH and the aetiology of MIH. The majority of PD (83.3%) can differentiate the MIH-related caries pattern from the classic caries pattern. Moreover, the confident level in diagnosing and treating MIH teeth were higher in PD compared to GD and DS (*p* = 0.000). The majority of dental practitioners in all groups agreed on the necessity to obtain continuing education on MIH including the aetiology, diagnosis, and its treatment to improve their knowledge and confidence in MIH clinical management.

## 1. Introduction

Molar incisor hypomineralisation (MIH) is a common developmental defect affecting from one to four first permanent molars and may involve permanent incisors. It is a qualitative defect of systemic origin [1]. The terms mottled enamel, non-endemic mottling of enamel, internal enamel hypoplasia, cheese molars, non-fluoride enamel opacities, opaque spots, and idiopathic enamel opacities were used to describe this defect [2]. Weerheijm initially introduced the MIH terminology in the European Association of Paediatric Dentistry meeting in 2001, which was later adopted globally [3]. MIH-like lesions can also be found in primary molars and are reported as predictors of MIH in permanent teeth [4,5,6,7]. Although the aetiologic factors of MIH remain unclear, it was suggested that MIH has a multifactorial model related to timing, strength, and duration of occurrence of the associated aetiological factors [3]. Recent systematic reviews report the involvement of genetic and environmental factors, such as acute and chronic illness during pregnancy through to the first three years of age in children as putative factors of MIH [3,8,9]. Hypoxia-related perinatal problems, certain infant and childhood illnesses and medication appear to increase the risk of having MIH [3]. Alterations in the function of the ameloblasts during the maturation phase may occur between the end of pregnancy and the age of 4 years and may lead to enamel hypomineralisation [3].

The clinical appearance of MIH ranges from mild to severe lesions. Mild lesions appear as white to brown demarcated opacities [3,10,11]. The darker colour of a lesion represents higher enamel porosities. This type of lesion tends to break down over time, leaving an opened dentin surface. Post-eruptive enamel breakdown and atypical caries are categorised as severe lesions [12,13,14]. MIH is characterised by asymmetrical lesions in location, size, and severity to other collateral teeth in the same patient [10,11]. Poor oral health due to hypersensitivity is commonly found in severe cases, and this condition makes MIH teeth prone to caries [15]. Patients with severe MIH lesions also experience difficulties with anaesthesia [16,17]. These conditions result in children with severe MIH having higher anxiety levels, thus making clinical management of MIH teeth a major challenge for dental practitioners [17,18].

Globally, the incidence of MIH is quite high, differs among regions and countries, with prevalence ranging from 2.2% to 44% [3,19,20,21,22]. Most of the studies about the prevalence of MIH are conducted in European countries; only few studies represent data from Southeast Asia [21,22,23,24,25]. There are limited data on MIH prevalence in Indonesia; however, dental caries’ prevalence in children remains high, which might be partially attributed to undiagnosed MIH [26]. Moreover, the number and distribution of paediatric dentists in Indonesia, who are trained to diagnose and treat any developmental tooth defect in children, are not yet sufficient. Therefore, general practitioners and other dental specialists might play an important role in diagnosing and managing MIH. The aim of this study was to evaluate perceptions and knowledge of MIH of general dental practitioners (GD), paediatric dentists (PD), and other dental specialists (DS) in Indonesia.

## 2. Materials and Methods

Ethical clearance was obtained from the Dental Research Ethics Committee, Faculty of Dentistry Universitas Indonesia. All respondents have consented to participate in this cross-sectional study by completing an online self-administered questionnaire anonymously and voluntarily using Google Forms. The questionnaire link was sent to the respective Indonesian Dental Association area coordinators, who in turn invited the dentist to participate in this study by distributing the information using social media such as Instagram, Facebook, and WhatsApp groups. The population of this study was GD, PD, and DS in Indonesia. The inclusion criteria were licensed dentists who provided dental services to children. No specific exclusion criteria were set. Data were collected for four weeks in November–December 2021. A modified questionnaire based on earlier publications was used [27,28,29,30,31,32]. Data on socio demography, knowledge, and confidence in diagnosing and treating MIH, preference for continuing education, and opinion about the MIH clinical training necessity were obtained. The questionnaire consisted of 3 parts. The first part contained questions related to sociodemographic data, such as age and gender in addition to qualification, and type–area–years of practice. The second part was about the knowledge of MIH and MIH awareness in terms of aetiology, prevalence, difference in caries patterns in MIH teeth, and time occurrence. The prevalence score was adjusted and modified due to the lack of available data in Indonesia to fit to a local context. The final section was about the perception and continuing education aspect of MIH.

Clinical photographs used in this study were used with consent, standardised, and were used in the MIH training manual for clinical fields surveys and practice [10]. Data were analysed using the IBM SPSS Statistics version 23.0 (SPSS Inc., Chicago, IL, USA) with a level of significance set at *p* < 0.05. Chi-squared tests were carried out for comparison between variables. Simple frequency distributions of sociodemographic variables in each group were tabulated and compared. Distribution and frequency tables were presented for descriptive analysis. Knowledge score (KS), a continuous variable, was computed based on previous studies [28,32]. Experts weighted and scored the answers about MIH knowledge [32]. The distribution of scores was agreed using Delphi methods for each question [32]. The KS for each respondent was obtained by summing the scores of all 10 questions (ranged from 20 to 60). Higher scores represented higher knowledge regarding MIH [28,32]. The Kruskal–Wallis test was used to compare the KS between the groups of respondents.

## 3. Results

The total number of respondents was 302. Ten GD and 26 DS were excluded because they did not provide dental services for children, leaving a total of 266 participants to be analysed. Table 1 showed the sociodemographic data of participants such as gender, age, and years–type–area of practice. Most of the respondents were female. The mean ages of GD, PD, and DS were 37.3 ± 9.9, 41.2 ± 8.8, and 41.8 ± 8.6 years old, respectively. The mean working experiences of GD, PD, and DS were 11.8 ± 9.1, 15.6 ± 8.8, and 14.1 ± 8.7 years, respectively. Most GD are working in a group practice, while PD and DS are mostly working in hospitals. The majority are working in urban areas, especially in the capital city. Most GD, PD, and DS agreed that there was an adequate training of MIH during their dental education.

Table 2 presented KS of MIH in GD, PD, and DS. The mean KS for all respondents was 46.9 ± 6.7, ranging from 23 to 60. There was statistically significant difference in the overall KS between GD, PD, and DS (*p* < 0.001). Most PD agree that fluoride exposure is not the aetiologic factor of MIH, contrary to most GD and DS (*p* < 0.032). Most PD were aware that the caries pattern related to MIH is different from the classical pattern (83.3%), compared to only 58% of GD and 63.3% of DS (*p* < 0.004).

Perception and continuing education aspects for GD, PD, and DS regarding MIH are presented in Table 3. There were significant differences between groups in their confidence in diagnosing MIH. More than half of PD felt confident or very confident in diagnosing MIH, while more than half of GD and DS felt unconfident or very unconfident. There were also significant differences between groups about their confidence in treating MIH. More than half of PD felt confident or very confident in treating MIH, while 63.2% GD and 58.4% DS felt unconfident or very unconfident. Most PD received information about MIH, while GD and DS received significantly less information. Most PD received information from dental journals and continued education events. Almost all dental practitioners would like to have further training regarding tooth hypomineralisation. GP responded the highest on the need to have training on tooth hypomineralization.

## 4. Discussion

This study was the first to investigate the knowledge and perceptions regarding MIH among dental practitioners in Indonesia. It focused on the knowledge gap to identify ways to improve information about the diagnosis, aetiology, and management of MIH in Indonesia. The respondent of this study consisted of dental practitioners who provided dental treatment for paediatric patients from fourteen different provinces in Indonesia. Dental practitioners in this study were categorised into three groups (GD, PD, and DS). Most dental practitioners in this study practice in urban areas, and all the paediatric dentists practice in urban areas. This might indicate an uneven distribution of dental practitioners in Indonesia.

The KS between dental practitioners included in this study were comparable. Similar findings were reported in a previous study [27,28]. Higher KS in PD have also been reported in an earlier study [28]. Moreover, PD might encounter MIH more commonly, attributed to higher exposure of MIH lesions, and have more clinical experience, therefore might be associated with higher KS. Almost all PD received more information about MIH compared to GD and DS. Similar results were reported in recent studies [28,29,30]. Almost half of PD received information about MIH from dental journals, compared to GD and DS, only one-third of whom received information about MIH from a dental journal. Information about MIH from continued education for all dental practitioners remained low. Although most respondents think that they received adequate training in dental school regarding MIH management; this was not reflected in the KS. These results may highlight the importance of obtaining information on MIH from continuing education for all dental practitioners to enhance their knowledge. A similar result was also reported, where PD and DS obtained more information about MIH in dental journals rather than via continuing education [30]. Thirty percent of the respondents disagree to have adequate training in MIH management. This might emphasize that the probability of MIH is still being misdiagnosed as caries. An increasing recognition of the burden of this common condition should be recognized. Dentists should be encouraged to regularly appraise the basic science and clinical MIH literature to ensure that they provide the best possible care for their patients.

Many studies on the prevalence of MIH report various results among countries and regions. One of the factors in the variety of results of MIH prevalence may be due to the unstandardised research methods and different age groups of research subjects in addition to differences in diagnostic criteria [33]. In 2003, EAPD launched the MIH criteria index, which was established to capture the clinical signs of MIH that could not be identified with the previous index [1,3,34]. MIH prevalence comparisons between countries using standardised diagnosis criteria may improve accuracy.

Although the exact aetiology and pathogenesis of MIH remains unclear, the last 10 years of research suggests that MIH is caused by multifactorial systemic events, including the involvement of genetic or epigenetic factors [8,9]. This is reflected in the respondents’ answers about the possible aetiologic factors of MIH. Almost all respondents chose more than one option, assuming that MIH is associated with multiple factors, similarly to previous studies. The latest systematic review reports demonstrate that chronic illness affecting mother and child also plays an important role in MIH aetiology [8]. Perinatal and postnatal aetiologic factors are considered as important factors in MIH development. Perinatal factors (prematurity, hypoxia, and caesarean section) and postnatal factors (urinary tract infection, measles, gastric disorders, otitis media, kidney diseases, bronchitis, asthma, and pneumonia) are significantly related with MIH occurrence [9]. Prenatal factors were also considered as contributing factors, but the latest systematic reviews demonstrate that there is no significant correlation between specific maternal diseases such as eclampsia, pre-eclampsia, medication during pregnancy, gestational hypertension, and maternal renal disease and MIH, and only unspecified maternal illness was found correlated with MIH in the latest meta-analyses [3,8,9]. The present study reveals that most of the dental practitioner respondents agree that genetic, environmental factors, acute medical conditions affecting mother or child, chronic medical conditions affecting mother and child, and antibiotic use and other medications are associated with the aetiology of MIH. The same responses were also reported in the latest studies [31,32]. Regarding fluoride exposure, PDs have better knowledge that fluoride exposure is not related to MIH. On the other hand, half of GPs and DS think fluoride exposure is related to MIH.

In terms of knowledge about caries patterns related to MIH, most PDs were able to differentiate the caries pattern related to MIH from the classic caries pattern, while fewer GP and DS knew of the different patterns. This study demonstrates that PDs have better knowledge regarding caries patterns related to MIH. Similar results were reported where PD knowledge was higher than GD, although the difference was not significant [28]. It is essential to be able to differentiate the MIH-related caries pattern from the classic pattern of caries to determine the appropriate management, since the management of MIH lesions is different from that of classic caries [13,35]. Increased knowledge about caries patterns related to MIH might influence the success rate treatment of MIH teeth.

PDs were more confident than GD and DS in diagnosing and treating MIH. More information and training exposure will improve their confidence in managing MIH teeth. The same result was reported by a previous study, where almost all PD felt confident in diagnosing and treating MIH teeth [28,30,36]. PDs were exposed to more information about MIH through dental journals and continuing education. Lack of information and clinical experience for GD and DS might influence their confidence both in diagnosing and treating MIH. Similar results were reported, where PD were more confident in diagnosing and treating MIH [28,30]. In this study, both groups conveyed the need for further training in the diagnosis, aetiology, and treatment of MIH. The same results have also been reported for other countries.

The findings of this survey establish baseline data on the knowledge and perceptions of MIH in Indonesia, although this online survey using the Google form may not be representative of dental practitioners for the whole of Indonesia. The current study employed a self-administered questionnaire distributed thru social media. Potential limitations might occur; consequently, the results of this study must be interpreted carefully. Selection bias and relatively low response rate might affect the representativeness of the study for dental practitioners in Indonesia. There might be a possibility of the overestimation of certain perceptions regarding MIH due to response bias, because the respondents may only represent those who have a positive tendency towards the study objectives. Despite all limitations, these results offer valuable information about the current knowledge and perceptions regarding MIH among Indonesian dental practitioners, because there are no prior studies on this topic in Indonesia. To highlight that, the consistent set-up between the current study and previous similar studies is one of the strengths of the paper because it enables comparisons between different regions over time. There were very limited data about the MIH prevalence in Southeast Asia, especially in Indonesia. Hence, there is a necessity to determine MIH prevalence among children in Indonesia as well as to explore the MIH distribution, severity, and the impact on the quality of life in Indonesian children. Further investigations to analyse the contribution of MIH to caries incidences in Indonesia due to the caries-prone structure of MIH teeth is also suggested.

## 5. Conclusions

The objective of this study was to evaluate the knowledge and perceptions of MIH of GD, PD, and DS in Indonesia. It can be concluded from the results that PD have a higher knowledge, perception, and confidence in diagnosing and treating MIH teeth compared to GD and DS. There is a need for dental care providers, especially GD and DS, to receive further training and continued education about MIH so that they can gain confidence in managing patients with MIH teeth. The dissemination of the latest information about MIH, especially to GD, being the primary dental care service providers in Indonesia, is needed to ensure that MIH is accurately diagnosed, and the appropriate treatment applied.

## Figures and Tables

**Table 1 dentistry-10-00190-t001:** Demographic characteristics of the study participants in Indonesia, categorised into general practitioners (GD), paediatric dentists (PD), and other speciality dentists (DS).

Characteristic	Total (n = 266) n (%)	GD (n = 122) n (%)	PD (n = 84) n (%)	DS (n = 60) n (%)
Gender				
Male	57 (21.4%)	25 (20.5%)	10 (11.9%)	22 (36.7%)
Female	209 (78.6%)	97(79.5%)	74 (88.1%)	38 (63.3%)
Age group				
≤30	45 (16.9%)	43 (35.2%)	1 (1.2%)	1 (1.7%)
31–40	118 (44.4%)	40 (32.8%)	49 (58.3%)	29 (48.3%)
41–50	67 (25.2%)	27 (22.1%)	19 (22.6%)	21 (35%)
≥51	36 (13.5%)	12 (9.8%)	15 (17.9%)	9 (15%)
Years of practice				
<5	48 (18%)	43 (35.2%)	3 (3.6%)	2 (3.3%)
6–10	58 (21.8%)	21 (17.2%)	23 (27.4%)	14 (23.3%)
11–20	94 (35.3%)	32 (26.2%)	37 (44.0%)	25 (41.7%)
21–30	55 (20.7%)	22 (18%)	17 (20.2%)	16 (26.7%)
>31	11 (4.1%)	4 (3.3%)	4 (4.8%)	3 (5%)
Type of practice				
Solo private practice	59 (22.2%)	35 (28.7%)	12 (14.3%)	12 (20.0%)
Group practice	120 (45.1%)	67 (54.9%)	23 (38.1%)	21 (35.0%)
Hospital	87 (32.7%)	20 (16.4%)	40 (47.6%)	27 (45.0%)
Location of practice				
Urban	257 (96.6%)	114 (93.4%)	84 (100%)	59 (98.3%)
Rural	9 (3.4%)	8 (6.6%)	0	1 (1.7%)
Province of practice				
Aceh	1	0	1	0
North Sumatra	2	1	0	1
Riau	2	2	0	0
South Sumatera	7	5	1	1
Lampung	1	1	0	0
Jakarta	161	65	54	42
Banten	32	15	9	8
West Java	42	20	16	6
Yogyakarta	6	5	0	1
East Java	1	0	1	0
Central Java	3	3	0	0
Bali	4	1	2	1
West Borneo	3	3	0	0
South Borneo	1	1	0	0
There was adequate training in dental school regarding MIH management				
Agree	186 (69.9%)	81 (66.4%)	62 (73.9%)	43 (71.7%)
Disagree	80 (30.1%)	41 (33.6%)	22 (26.2%)	17 (28.3%)

**Table 2 dentistry-10-00190-t002:** Percentage distribution of MIH knowledge scores of general practitioners (GD), paediatric dentists (PD), and other speciality dentists (DS) in Indonesia (N = 266).

Knowledge Questions	Knowledge Scores		Percentage Distribution of Dental Practitioners Answered “YES” n (%)				*p*-Value
	Yes	No	All (n = 266) n (%)	GD (n = 122) n (%)	PD (n = 84) n (%)	DS (n = 60) n (%)	
Have you been aware that MIH is a developmental defect that differs from hypoplasia and fluorosis?	9	0	226 (85%)	102 (83.6%)	70 (83.3%)	54 (90%)	0.463
What is your opinion about the prevalence of MIH in your community? (One option chosen)							0.198
<5%	0	^	55 (20.7%)	28 (23.0%)	9 (10.7%)	18 (30.0%)	
5–10%	1	^	53 (19.9%)	24 (19.7%)	20 (23.8%)	9 (15.0%)	
10–20%	1	^	23 (6.8%)	11 (9.0%)	6 (7.1%)	6 (10.0%)	
>20%	1	^	24 (9.0%)	12 (9.8%)	9 (10.7%)	3 (5.0%)	
No national data available	6	^	111 (41.7%)	47 (38.5%)	40 (47.6%)	24(40.0%)	
Do you think these are MIH aetiologic factors?							
Genetics	5	4	191 (71.8%)	91 (74.6%)	55 (65.5%)	45 (75.0%)	0.296
Contaminants from environmental factors	5	4	186 (69.9%)	84 (68.9%)	61 (72.6%)	41 (68.3%)	0.807
Chronic medical conditions affecting mother and child	6	3	226 (85.0%)	98 (80.3%)	76 (90.5%)	52 (86.7%)	0.123
Acute medical conditions affecting mother/ child	6	3	163 (61.3%)	69 (56.6%)	57 (67.9%)	37 (61.7%)	0.262
Medications such as antibiotics	5	4	186 (69.9%)	85 (69.7%)	57 (67.9%)	44 (73.3%)	0.776
Exposure of fluoride	1	8	113 (42.5%)	57 (46.7%)	26 (31.0%)	30 (50.0%)	0.032 *
When do you think the insult occurs? (One option chosen)							0.616
During pregnancy	1	^	66 (24.8%)	36 (29.5%)	14 (16.7%)	16 (26.7%)	
First year of life	3	^	29 (10.9%)	12 (9.8%)	12 (14.3%)	5 (8.3%)	
Third year of life	0	^	41 (15.4%)	17 (13.9%)	16 (19.0%)	8 (13.3%)	
Pregnancy to first year of life	3	^	53 (19.9%)	22 (18.0%)	18 (21.4%)	13 (21.7%)	
Pregnancy to third year of life	2	^	77 (28.9%)	35 (28.7%)	24 (28.6%)	18 (30.0%)	
Do you think MIH related caries pattern differentiate from classical dental caries pattern?	7	1	179 (67.3%)	71 (58.2%)	70 (83.3%)	38 (63.3%)	0.004 *
Mean knowledge score (SD)			46.9 (6.7)	45.4 (6.3)	49.2 (7.1)	46.5 (6.3)	0.001
Range	Min. 20	Max. 60	23–60	23–59	33–59	36–60	

^ Answer “No” does not apply, as it was analysed as a single choice question. * Statistically significant (*p*-value < 0.05); Pearson’s chi-square test.

**Table 3 dentistry-10-00190-t003:** Confidence and continuing education aspects regarding MIH among general practitioners (GD), paediatric dentists (PD), and other speciality dentists (DS) in Indonesia.

Questions	All	GD	PD	DS	*p*-Value
Confidence in diagnosing and treating MIH					0.000 *
How do you feel about diagnosing MIH?					
Very confident	8 (3.0%)	1 (0.8%)	5 (6.0%)	2 (3.3%)	
Confident	128 (48.1%)	46 (37.7%)	56 (66.7%)	26 (43.3%)	
Unconfident	125 (47.0%)	74 (60.7%)	23 (24.7%)	28 (46.7%)	
Very unconfident	5 (1.9%)	1 (0.8%)	0	4 (6.7%)	
How do you feel about treating MIH?					0.020 *
Very confident	5 (1.9%)	1 (0.8%)	2 (2.4%)	2 (3.3%)	
Confident	114 (42.9%)	44 (36.1%)	47 (56.0%)	23 (38.3%)	
Unconfident	126 (47.4%)	64 (52.5%)	34 (40.5%)	28 (46.7%)	
Very unconfident	21 (7.9%)	13 (10.7%)	1 (1.2%)	7 (11.7%)	
**Continuing education**					0.001 *
Are you receiving any information on MIH? (YES)	217 (81.6%)	96 (78.7%)	79 (94.0%)	42 (70.0%)	
Books	7 (2.6%)	1 (0.8%)	4 (4.8%)	2 (3.3%)	
Campus	41 (15.4%)	22 (18.0%)	11 (13.1%)	8 (13.3%)	
Continuing Education	44 (16.5%)	15 (12.3%)	21 (25.0%)	8 (13.3%)	
Dental journal	100 (37.6%)	41 (33.6%)	40 (47.6%)	19 (31.7%)	
Instagram	4 (1.5%)	2 (1.6%)	1 (1.2%)	1 (1.7%)	
YouTube	9 (3.4%)	6 (4.9%)	1 (1.2%)	2 (3.3%)	
Others	15 (5.6%)	9 (7.4%)	2 (2.4%)	4 (6.7%)	
Would you like further training regarding tooth hypomineralisation? (Yes)					
Diagnosis	240 (90.2%)	118 (96.7%)	75 (89.3%)	47 (78.3%)	0.000 *
Aetiology	235 (88.3%)	114 (93.4%)	76 (90.5%)	45 (75.0%)	0.001 *
Treatment	247 (92.9%)	119 (97.5%)	80 (95.2%)	48 (80.0%)	0.000 *

* Statistically significant (*p*-value < 0.05); Pearson’s chi-square test.

## Data Availability

The raw data are available from the authors to any author who wishes to collaborate with us.

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
