# Peer review of "Knowledge and Perceptions of Molar Incisor Hypomineralisation among General Dental Practitioners, Paediatric Dentists, and Other Dental Specialists in Indonesia"

_dentistry, 2022, doi:10.3390/dj10100190_

Round 1

Reviewer 1 Report

The paper overall is well presented and has good methodology. There are some information that needs to be clarified in the materials and method sections that would improve the paper. However, I am skeptical in regards to the clinical significance of this paper since many similar studies have been conducted globally. Maybe a more local journal would be more appropriate to present the data for Indonesia. 

Some points to consider for improvement: 

Introduction: The last paragraph needs some syntax editing to make sense. Also, I am not sure the term multifactorial is appropriate for MIH. We usually use it in caries as there are multiple factors that need to be present for its formation. With MIH something disturbs the tooth formation but this doesn't mean that there are multiple factors necessarily. I would stay to the facts saying that the etiology is unclear and these factors have been related with MIH occurrence. 

Materials & Method: 1) To whom were the online surveys distributed through the social media? Were these forums of dentists, dental school alumni? This information would be important to understand how representative was the sample. According to the authors in the limitations presented the sample was not representative and the response rate was low but this information is not presented in the materials and methods. 2) Scoring details need to be analyzed further as they are somehow presented in Table 2 but is not clear how the authors decided to score one question with 9 points other answers with 6 etc. 

Discussion: The paragraph (3rd) of MIH prevalence in Indonesia should be omitted as it does not present any data related to this study. Some of this information can be added to the end of the discussion where authors state the needs for future studies on MIH. 

Reviewer 2 Report

very interesting study.

Authors must explain why they chose GD, PD and other specialties.

Obviously, PD are the ones who see MIH more commonly so they have more knowledge. (Justify in discussion)

the below information was missing:

Photographs were used in the study and what was the reference point for the details 

All the photographs were standardized

SPSS used for analysis 

No clear information on  demographic details in the results (only mentioned in the tables )

30% said they don't have knowledge of MIH 

Authors should justify this 

30% of the participants were accepted that they don't have sufficient training how they are able to give appropriate feedback 

This is always questionable.

Inclusion and exclusion criteria is not clear.

 Gamboa et al Reference was not stated.

Google form was sent through social media always questionable. (Justify in discussion)

Limitations and strengths of the study are missing.

Concise the conclusion and keep as objective-based  

Reviewer 3 Report

Dear Authors,

the manuscript "Knowledge and Perceptions of Molar Incisor Hypomineralisation among General Dental Practitioners, Paediatric Dentists, and Other Dental Specialists in Indonesia" was interested in reading. however, there were some comments. please go through the attached PDF

Round 2

Reviewer 2 Report

MIH prevalence may be due to the unstandardised research methods and different age groups of research subjects in addition to differences in diagnostic criteria [31,32]. .................... Remove reference No 32 here 

In 2003, EAPD launched the MIH criteria index which was established to capture the clinical signs of MIH that could not be identified with the previous index [32]. ...............add references No 1 and 3  here 

MIH is caused by multifactorial systemic events, including the involvement of genetic or epigenetic factors [3,8,9]. ..............remove Reference No three here

Add a questionnaire as a supplementary document. 

Conclusion: Should be based on Objective 

Author Response

Response to reviewer 2 (2nd round):

Thank you very much for the valuable feedback.

Point 1: MIH prevalence may be due to the unstandardised research methods and different age groups of research subjects in addition to differences in diagnostic criteria [31,32]. .................... Remove reference No 32 here

Response 1: Reference was removed as suggested.

Point 2: In 2003, EAPD launched the MIH criteria index which was established to capture the clinical signs of MIH that could not be identified with the previous index [32]. ...............add references No 1 and 3 here

Response 2: References were added accordingly.

Point 3: MIH is caused by multifactorial systemic events, including the involvement of genetic or epigenetic factors [3,8,9]. ..............remove Reference No three here

Response 3: Reference was removed.

Point 4: Add a questionnaire as a supplementary document.

Response 4: The questionnaire was uploaded as a supplementary document.

Point 5: Conclusion: Should be based on Objective

Response 5: Conclusion was revised.

Round 3

Reviewer 2 Report

The manuscript is better than previous versions.

The authors addressed all the queries.